# Preparation of Natural Plant Polyphenol Catechin Film for Structural Coloration of Silk Fabrics

**DOI:** 10.3390/biomimetics9010015

**Published:** 2024-01-01

**Authors:** Shuaikang Yang, Desheng Sha, Yijiang Li, Meiqi Wang, Xiaowei Zhu, Xiangrong Wang, Guoqiang Chen, Yichen Li, Tieling Xing

**Affiliations:** College of Textile and Clothing Engineering, China National Textile and Apparel Council Key Laboratory of Natural Dyes, Soochow University, Suzhou 215123, China; 2015404043@stu.suda.edu.cn (S.Y.); 20214215016@stu.suda.edu.cn (D.S.); szdxleeyj@163.com (Y.L.); 2015404068@stu.suda.edu.cn (M.W.); xiaoweiz125@163.com (X.Z.); wangxiangrong@suda.edu.cn (X.W.); chenguojiang@suda.edu.cn (G.C.)

**Keywords:** catechin, structural color, silk fabric, film interference

## Abstract

Traditional textile dyeing uses chemical pigments and dyes, which consumes a large amount of water and causes serious environmental pollution. Structural color is an essential means of achieving green dyeing of textiles, and thin-film interference is one of the principles of structural coloring. In the assembly of structural color films, it is necessary to introduce dark materials to suppress light scattering and improve the brightness of the fabric. In this study, the conditions for the generation of nanofilms of catechin (CC) at the gas–liquid interface were successfully investigated. At the same time, environmentally friendly colored silk fabrics were novelly prepared using polycatechin (PCC) structural color films. In addition, it was found that various structural colors were obtained on the surface of silk fabrics by adjusting the time. Meanwhile, the color fastness of the structural colored fabrics was improved by introducing polyvinylpyrrolidone (PVP) to form a strong hydrogen bond between the fabric and catechin. PCC film is uniform and smooth, with a special double-layer structure, and can be attached to the surface of silk fabrics, giving the fabrics special structural colors. Through the thin-film interference formed between the visible light and the PCC film, the silk fabrics obtain bright, controllable, and uniform structural colors. This method is easy to operate and provides a new way of thinking for environmental-protection-oriented coloring of fabrics.

## 1. Introduction

The allure of nature is inextricably linked to its vivid array of colors. Blue sky, white clouds, sunsets, peacock feathers, and insect shells are all involved in the wonderful science of color. Within the natural world, colors can be divided into pigment colors and structural colors according to their generation mechanisms. Pigment colors arise from the selective absorption and reflection of visible light by electrons in the orbitals or energy bands of dye molecules [1,2]. Structural colors, on the other hand, are colored due to the physical optics of diffraction, interference, or scattering of light by their unique periodic physical micro-nanostructures. Nowadays, dyes and pigments find extensive applications in textile printing and dyeing processing. Nevertheless, the usual chemical dyeing process results in a large amount of energy consumption and the generation of dyeing wastewater, which poses a severe threat to environmental integrity [3,4]. Furthermore, textiles dyed with chemical colorants are susceptible to fading over time. To address these pressing issues, developing an eco-friendly and colorfast textile dyeing technology holds profound significance.

Structural color technology is a physical coloring technique, which uses special structured color material to produce light scattering, diffraction, interference, and other optical effects, resulting in a variety of bright colors. In contrast to conventional dyes and pigments, structural colors usually have the characteristics of high brightness, exceptional saturation, and non-photofading [5,6]. While structural coloring still faces challenges, such as low durability, high production costs, and difficulty in large-scale preparation, the use of structural colors in textile dyeing and the development of new textile dyeing technologies have garnered significant attention. Thin-film interference is one of the main principles of structural color, which can be applied to textile structural coloration. This entails the construction of a film layer with distinctive optical properties on the surface of fabric substrates. Figure 1 shows the principle of thin-film interference to produce structural colors. When light (L_1_) strikes the surface of the film, the lights (R_11_, R_22_ (R_11_, R_22_ are the parallel rays of R1 and R2 respectively.)) will converge on the surface of the film through the refraction (R_2_) and reflection (R_1_) of the film, generating a specific wavelength of visible light, thus bestowing vibrant structural colors upon the fabric [7,8]. Notably, the structural color can be finely tuned by manipulating the structural parameters of the film, including its thickness and refractive index, among others [9,10].

The existing structural color technology still faces several challenges. The color of structurally colored fabrics is often compromised by ambient light scattering and incoherent light, reducing saturation and introducing partial whiteness. To mitigate these issues, dark materials are frequently introduced to suppress the light scattering [11,12]. In the assembly of structural color films, substances like carbon black [13] and polydopamine [14] have been employed. Nonetheless, the direct incorporation of black material can induce uneven color due to physical deposition [15], and the utilization of dopamine is hindered by its high cost [16]. Moreover, the bonding strength between the structural color film and fabric substrate remains weak, primarily relying on a limited presence of hydrogen bonds and van der Waals forces [17]. The capillary effect generated during solvent evaporation induces transverse tensile stresses, ultimately forming cracks in the film [18,19].

Phenolic and polyphenolic compounds, widely distributed in plant tissues, serve diverse biological functions, including chemical defense, pigmentation, structural support, and radiation damage prevention [20,21]. Among these, catechin (CC), a crucial plant polyphenol, possesses a distinctive catechol structure that imparts the abilities of oxidation and polymerization. Additionally, catechins exhibit dopamine-like properties that can form nanofilms at the gas–liquid interface. New aggregates form constantly, gradually increasing in size through integration with monomeric species, free oligomers, and other aggregates during extended polymerization periods [22]. The resultant aggregated particles demonstrate hydrophobic characteristics, causing them to migrate towards the surface of the liquid. However, as the particles grow and the force of gravity is greater than the hydrophobic force, they descend to the bottom of the solution. As the number of particles in the upper layer increases, a nanofilm emerges. When the thickness of the film reaches a certain thickness, it refracts the visible light, resulting in the production of structural color. To expedite the polymerization time of polyphenols, Yang et al. [23] used CuSO_4_ as an oxidizing agent and H_2_O_2_ as a trigger to promote the rapid oxidation and deposition of caffeic acid, and obtained a uniform, stable, and multifunctional dark brown polycaffeic acid surface coating. Notably, this catalytic process also proves suitable for the oxidative polymerization of CC. The phenol hydroxyl structure of a CC molecule can form a ring chelate by complexing with more than two coordination atoms of a metal ion [24], thereby enhancing the cross-linking degree of the system. Furthermore, the CC and metal ions complex exhibits a yellow–brown color, effectively reducing the influence of white scattered light on the saturation of structural colors. Additionally, the catechol structure of CC endows it with a robust interfacial force, allowing direct adherence to any substrate surface [25]. Simultaneously, introducing hydrophilic polymer PVP into the system forms strong hydrogen bonds with polyphenol compounds, which can inhibit the cracks induced by surface tension to a certain extent, further improving the structural stability of the film. Consequently, polycatechin films (PCC) hold significant potential for achieving structural color on textile surfaces.

In this work, plant polyphenol (catechins) was employed to construct structural color films, and this novel approach has not been previously investigated. The application of this technique in textile coloring was also performed. Catechins served as the primary raw material to construct PCC structural color films on the surface of pristine white silk fabrics through the catalytic oxidation of metal ions Fe^3+^, Cu^2+^, and hydrogen peroxide to obtain bright structurally colored fabric. The stepwise preparation procedure is illustrated in Figure 2. Additionally, PVP was incorporated into the PCC film to establish robust hydrogen bonds, thereby enhancing the mechanical stability of the film. Notably, this approach boasts simplicity in execution and utilizes eco-friendly raw materials, offering a novel avenue in the field of textile structural coloration technology.

## 2. Experimental

### 2.1. Materials

The degummed silk fabric (36 g/m^2^) was provided by Huajia Silk Group Co. Ltd., Suzhou, China. Catechin (CC) and tris-(hydroxymethyl) aminomethane (Tris) were purchased from Sangon Biotech Co., Ltd., Shanghai, China. Cupric chloride anhydrous (CuCl_2_, 99%), ferric chloride (FeCl_3_, 99%), and hydrogen peroxide (H_2_O_2_, 30%) were bought from Sinopharm Chemical Reagent Co., Ltd., Shanghai, China. PVP (M_w_ = 36 kDa) was supplied by Sigma-Aldrich Co., Ltd., Shanghai, China. All the chemicals were of analytical reagent grade and used without further purification.

### 2.2. Experimental Section

#### 2.2.1. Preparation of S-PCC-n

CC (0.03 mol/L) was dissolved in Tris (pH = 7.8, 50 mmol/L) with magnetic stirring in a water bath at 45 °C. CuCl_2_ (6 mmol/L), FeCl_3_ (1.2 mmol/L) and H_2_O_2_ (17.6 mmol/L) were added and stirred well. It is configured as the precursor reaction liquid of PCC structure color film. A round silk fabric (degummed) with a diameter of about 5 cm was selected and placed in a Petri dish after drying, and 8 mL of precursor reaction liquid was added. The reaction was controlled at 25 °C (room temperature) at different times. After a period of reaction, the excess reaction liquid was absorbed. The treated sample was dried at a constant temperature (25 °C) to obtain silk fabric with catechin structure color film. The above structural colored silk fabric was marked as S-PCC-n, where S represents the silk fabric and n represents the polymerization time of PCC.

#### 2.2.2. Preparation of S-PCC-mPVP-n

Moreover, 5 mL PVP solution of different concentrations was added to the structure color fabric in the Petri dish. After 2 h of reaction, the excess solution was sucked out and dried at 30 °C until the fabric quality no longer changed. Thus, the PVP-treated PCC structure color silk fabric was obtained. The above structural colored silk fabric was marked as S-PCC-m PVP-n, where m indicates various PVP concentrations, including 5, 10, and 20 mg/mL, and n indicates the polymerization time of CC.

### 2.3. Characterization

The surface topology of PCC structural colored silk fabric was studied by Atomic Force Microscopy (Bruker, Billerica, MA, USA) in a tapping mode with a scanning range of 2 μm × 2 μm. The chemical structure of the structural colored silk fabric was tested by Fourier Transform Infrared Spectroscopy (FTIR) at a scanning range of 500–3500 cm^−1^. The surface elemental content of the samples was analyzed by X-ray photoelectron spectroscopy (XPS, Thermo Scientific ESCALAB Xi+, USA) using Al Ka excitation radiation (1484.6 eV). The crystal structures of samples were characterized using an X-ray diffractometer (XRD, Rigaku Ultima IV Japan) with Cu Kα (λ = 1.54 Å) as the radiation source in the range of 10 to 80°, a supply voltage of 40 kV, and a scanning speed of 5°/min. The types and contents of component elements in PCC nanofilm were analyzed by an Energy Dispersive Spectrometer (EDS, Hitachi Regulus 8230, Japan). The morphological characterizations of the samples were analyzed by field emission scanning electron microscope (SEM, Hitachi Regulus 8230, Japan) with the accelerating voltage maintained at 5.0 kV. The reflection spectra were measured by UV–Vis-NIR Spectrophotometer (UV3600, Japan). The K/S values of structural colored silk fabric were determined by reflection spectrophotometer (UltraScan PRO, USA), and the average of four parallel measurements was obtained under the D65 light source and 10° observation angle measurement conditions. The color fastness of structurally colored fabric was examined by sandpaper abrasion test [26] and soaping test. The sandpaper abrasion test was carried out as follows: sandpaper adhered to the bottom of the 100 g weight, and then it was pulled on the structural color film of the fabric surface 10 times to observe the change in structural color. The soaping test was performed by washing the structurally colored silk fabrics in a soap solution at a 1:50 bath ratio for 30 min at 40 °C and repeated three times. Refractive index test: Refractive Index Testing of Samples by spectroscopic ellipsometry (America, J.A.Woollam IR-VASE Mark II M-2000UI); five points were selected for sampling and testing, and the evaluated values of the tests were taken. Sample preparation: sample preparation by reference to the preparation method of PDA film [27,28].

FDTD simulation: in this paper, the reflectance spectra of PCC films with different thicknesses are simulated by the finite difference time domain (FDTD) method using the software Lumercial 2020a Finite Difference IDE. All simulations are executed at a spatial resolution of 0.5 μm while incorporating periodic boundary conditions in both the x- and y-directions, along with a perfectly matched layer (PML) absorber in the z-direction. The incident wave source consisted of visible light propagating backward along the z-direction in the wavelength range of 350–750 nm. Notably, the film thickness was adjusted based on the experimental results and continuously optimized to closely match the measured reflectance spectrum. The thickness of PCC film was ascertained through an analysis of cross-sectional SEM images. We have added the above details of the simulation into the revised manuscript.

## 3. Results and Discussion

### 3.1. Color Performance Characterization

In this investigation, the structural colors predominantly arise from the interference occurring between the incident light and the thin-film structure [29,30]. The photographs of structurally colored silk fabrics under different reaction times are displayed in Figure 3. The structural color of silk fabrics changed with the increase in dopamine polymerization time. As shown in Figure 3a, the color on the silk fabric changed from the original white to yellow after 1.5 h of polymerization. Subsequently, at 2.5 h, 4 h, 6 h, and 9 h, the structural colors evolved into shades of purple, blue, green, and red, respectively. From the optical micrograph of the silk fabric in Figure 3f–j, it can be seen that the surface of the fabric is covered with a film, which interacts with light, resulting in diverse structural colors on the silk fabrics. Remarkably, these structural colors exhibit continuous changes in response to the extension of the reaction time, which can be attributed to the gradual thickening of the PCC film as the reaction time progresses. However, it is worth noting that the structural colors on the fabrics exhibit slight non-uniformity, which may be caused by the uneven distribution of active groups and the presence of fiber irregularities within the fabrics, leading to varying growth rates in PCC film at different locations [31].

Figure 4a shows the reflection spectrum of the structural colors on the silk fabrics. Different structural colors on the surface of silk fabrics exhibit unique reflectance curves. Notably, in the case of yellow silk fabric, reflectance gradually decreased to 480 nm, after which it underwent a gradual increment. For a polymerization duration of 2.5 h, reflectance decreased in the range of 400~580 nm, subsequently rising as wavelength surpasses 580 nm. The surface reflectance curve of S-PCC-4 h fabric corresponds to a blue reflectance peak at 440 nm, which appears blue. The colored silk fabric reacting for 6 h exhibits a prominent green reflection peak at 530 nm. The S-PCC-9 h fabric has two reflection peaks at 460 nm and 630 nm, corresponding to the colors of blue and red, respectively. The heightened reflectance of red light imparts a pink hue to the fabric. The results of reflection spectra show that the color of the structure changes with the polymerization period, and the maximum reflection peak shifts to a longer wavelength with the increase in the polymerization period. This confirms that the coloration of silk fabrics can be effectively achieved by modulating the self-polymerization duration of CC, offering a viable alternative to conventional chemical dyes. K/S value is the ratio of color absorption coefficient K to color scattering coefficient S, commonly used to indicate the color depth of the surface of a solid sample [32]. As shown in Figure 4b, the K/S value of the structure color of silk fabric is obviously different with different PCC polymerization times.

To further observe the structurally colored silk fabrics formed under different CC polymerization times, the above reflection spectra were converted into CIE chromaticity coordinates. As depicted in Figure 4c,d, the shift in CIE chromaticity coordinates from yellow to pink aligns seamlessly with the visual representation presented in Figure 3. Consequently, various colors can be controllable on the surface of silk fabrics via CC polymerization.

### 3.2. Microstructure Characterization of PCC Films

The mechanism of color rendering on the surface of silk fabric originates from the unique physical structure of PCC films. As shown in Figure 5, different structures and morphologies on the upper and lower layers of PCC film on the surface of silk fabric were observed by scanning electron microscopy (SEM). The upper layer of PCC film is a dense film layer, and the accumulation of particles forms the bottom layer. The thickness of a film determines the optical range difference experienced by light passing through the film. When a film reaches a certain thickness and light is incident vertically from the air onto the surface of the film, part of the light is reflected, and part of the light is transmitted. These two parts of the light are reflected and refracted many times inside the film and then meet again, forming an interference phenomenon. When the difference in optical range is equal to an integer multiple of the wavelength of the light, the interference phenomenon is enhanced and a bright structural color is formed. As the thickness of the film varies, the resulting optical range difference changes accordingly, resulting in different structural colors. It can be seen from the cross-section of PCC film that, after polymerization for 1.5 h, 2.5 h, 4 h, 6 h, and 9 h, the thickness of the upper film is 114.5 nm, 179.5 nm, 237.5 nm, 274.6 nm, and 355.6 nm, respectively, as shown in Table 1. The thickness of the upper layer increased with the extension of polymerization time, and the specific wavelength of the incident light produced by interference changed, which became the main factor affecting the color of the fabric. The lower layer is deposited as a dark material, which can absorb irrelevant scattered light in the film, enhancing the vividness and saturation of the structural color, as shown in Figure 3f.

The microscopic surface morphology of PCC film silk fabric was further characterized by atomic force microscopy (AFM), as shown in Figure 6. The average roughness of PCC film on the fabric is 4.44 nm, indicating a smooth and continuous surface of PCC film. Therefore, the method of preparing structural color film with CC has a better film-forming property, uniform film thickness distribution, and high color uniformity.

### 3.3. Surface Composition Analysis of Silk Fabric Modified with PCC

The surface structure of silk fabric modified with PCC was analyzed. As shown in Figure 7a,b, the structure of PCC film is mainly composed of four elements: C, O, Cu, and Fe, among which C and O elements are from CC, and the presence of Cu and Fe elements proves that Cu^2+^ and Fe^3+^ are involved in the composition of PCC film. Figure 7c,f show the distribution of C, O, Cu, and Fe, respectively, indicating that these four elements are evenly distributed in the film. The oxidative polymerization reaction of catechins is very complex and is generally considered to be a free radical process, whereas the Cu^2+^/H_2_O_2_ system is able to generate a large number of reactive oxygen radicals (•OH, O_2_^−^•, and HO_2_•, etc.), which are capable of inducing oxidation. On the other hand, the oxidation reduction potential of catechins to quinone-type is much lower than that of Cu^2+^, which can oxidize catechins. At the same time, the introduction of trace amounts of Fe^3+^ blackens the solution, which not only eliminates the effect of the color of catechin itself on structural colors but also provides a black background for the structural colors, resulting in a certain increase in color saturation. Based on the aforementioned reasons, two metal ions, Cu^2+^ and Fe^3+^, were added to the system to enhance the saturation of the structural color while ensuring rapid formation of PCC structural color film.

The crystal structures of the pristine silk fabric and S-PCC-4 h were studied by X-ray diffraction (XRD). The XRD curves of the two samples are similar, and the characteristic diffraction peaks at 20.5° and 24.4° are observed on the XRD pattern, as shown in Figure 8a. The diffraction peak at 20.5° comes from the β-sheet structure of silk, and the diffraction peak at 24.4° may come from the α-helix and β-sheet structure of silk [32,33]. The above results show no significant difference in the crystal structure between the pristine ilk fabric and PCC-modified silk fabric. In other words, the molecular structure of PCC polymer is dominated by amorphous regions.

The FTIR spectra showed the difference among functional groups on the surface of silk fabric before and after being modified with PCC structural color film, as shown in Figure 8b. The absorption peaks of the pristine silk fabric at 1622 cm^−1^ and 1510 cm^−1^ are generated by C=O stretching vibrations and in-plane bending of N-H, respectively. For S-PCC-6 h, the peak at 3277 cm^−1^ is caused by the stretching vibration of -OH [34,35], while the peaks at 814 cm^−1^ are attributed to substituted phenyl [36]. The -OH increase was not insignificant, indicating that some -OH of catechin participated in the complexation with Cu^2+^ and Fe^3+^. The peak around 1058 cm^−1^ is usually associated with phenol C-O stretching vibrations, while the 1220 cm^−1^ peak comes from O-H in-plane bending. The new peak at 2845 cm^−1^ may come from the C-H symmetric stretching vibration [37].

Figure 8c shows the XPS spectra of the pristine silk fabric and S-PCC-4 h. Compared with the original silk fabric, the content of O element on the surface of S-PCC-4 h is significantly increased due to the large amount of phenol hydroxyl carried by catechins. It also contains small amounts of Cu and Fe elements. In addition, the high-resolution C1s spectrum of S-PCC-4 h shows three distinct peaks (Figure 8d). The characteristic peaks of 288.3 eV, 286.4 eV, 286.1 eV, and 284.8 eV represent C=O, C-OH, C-N/C-O, and C-C [38], which are derived from the ester group, phenolic hydroxyl group, and aliphatic ring of PCC, respectively. The above results prove that PCC film has successfully attached to the surface of silk fabric.

### 3.4. Mechanical Stability of PCC Films

The mechanical stability of PCC films determines the color fastness of structurally colored silk fabrics. In order to test the mechanical stability of PCC film, the PCC-modified silk fabric was first subjected to 30 repeated folding tests of PCC, as shown in Figure 9a. Figure 9b,c show the electron microscope images of the surface of PCC silk fabric before folding and the crease of PCC-modified silk fabric after folding. It can be seen that the PCC film at the crease does not fall off and break, which proves that the PCC film has good flexibility on the surface of silk fabric.

In order to test the anti-abrasion performance of PCC films and the influence of PVP on the mechanical stability of PCC films, abrasion tests were carried out on PCC-modified silk fabrics and PCC-modified silk fabrics treated with different concentrations of PVP. As can be seen from Figure 10a, the upper film of the structural colored fabric without PVP completely fell off after abrasion, exposing the accumulation of the lower dark polymer and losing the original color. Compared with the structurally colored fabric without PVP, the surface color and structure of the structurally colored fabric with PVP treatment are better maintained after abrasion. Combined with the reflection spectrum of the fabric in Figure 10b,c, it can be found that the reflection curve of S-PCC-4 h after abrasion shows the most significant change, followed by S-PCC-5 mg/mLPVP-4 h. However, the reflection curves of S-PCC-10 mg/mLPVP-4 h and S-PCC-20 mg/mLPVP-4 h show little change after abrasion. This may be due to PVP acting as a strong hydrogen acceptor, utilizing its γ-lactam to form a strong hydrogen bond with the catechol group on PCC [39], which improves the internal force of the film. In addition, it can be found from Figure 10b that the reflectance of the structural colored silk fabric gradually decreases with the increase in PVP concentration. Excessive PVP will increase the light absorption of film, resulting in a decrease in the color saturation of silk fabric. Consequently, it was concluded that 10 mg/mL is more appropriate.

In order to further detect the influence of PVP on the structural color stability, a soaping test was performed on S-PCC-4 h and S-PCC-10 mg/mL PVP-4 h. It can be seen from Figure 11 that S-PCC-4 h exhibits obvious decolorization after washing, and the reflection curve also undergoes significant changes, basically losing the original color. Similarly, S-PCC-10 mg/mL PVP-4 h after washing also faded, but some of the original color was still retained, and the reflectance decreased, but the shape of the reflection curve still existed. The results show that introducing PVP can slightly improve the washing fastness of PCC structural colored silk fabric.

As can be seen from Figure 12a, the surface of untreated silk fabric is smooth and flat. Figure 12b,c shows that the structural color film on the surface of fabrics with PVP is more complete. Figure 12d,e show the SEM images of the structural color fabrics after three soaping operations. Indeed, the surface film of the structured fabric without PVP has been completely detached, and only some nanoparticles are attached to the silk surface. On the other hand, only part of the structural color film of the S-PCC-10 mg/mLPVP-4 h has been detached. The addition of PVP helps to improve the fastness of the structural color film.

Figure 13 shows the refractive index profile of the PCC film in the range of 350–1600 nm. Figure 14 shows a picture of the simulation model as viewed from different viewpoints. Figure 15 shows the comparison data between the fitted spectra and the actual measured spectra. The data fitted from Figure 15 are in general agreement with the actual measured spectral data, which effectively proves that the substance is capable of producing structural colors, whereas the spectra fitted in Figure 15d,e differ from the actual measured spectra. This may be due to the unevenness of the silk fabric surface undulation and uneven distribution of functional groups caused by the uneven deposition of the film, as well as due to the action of the liquid surface tension caused by film cracking. These reasons led to a slight unevenness and inaccuracy in color of the prepared structured color fabrics. Figure 15f shows the refractive index of the material during the simulation, which is consistent with the data shown in Figure 13.

## 4. Conclusions

In this study, the conditions for the generation of PCC nanofilms at the gas–liquid interface have been successfully investigated, and PCC nanofilms were also applied to textile coloring for the first time. This PCC film exhibits a dual-layered structure consisting of an upper film and lower particles. The upper layer of PCC produces structural color via light interference, while the lower particles absorb irrelevant scattered light, thereby increasing the saturation of structural color. As the polymerization duration prolongs, the thickness of the PCC film gradually increases, yielding a diverse array of structural colors. Specifically, yellow, purple, blue, green, and pink colors can be obtained after 1.5 h, 2.5 h, 4 h, 6 h, and 9 h, respectively. The results show that the color of PCC structure coloring fabric is bright, controllable, uniform, and continuous. After introducing PVP, the PCC structure-colored fabrics manifest commendable mechanical stability. This study demonstrated the potential of CC in fabricating structural color films, established a foundation for advancing high-fastness structural color fabrics based on natural plant polyphenols, and advanced the practical application of structural color technology in textile coloring.

## Figures and Tables

**Figure 1 biomimetics-09-00015-f001:**
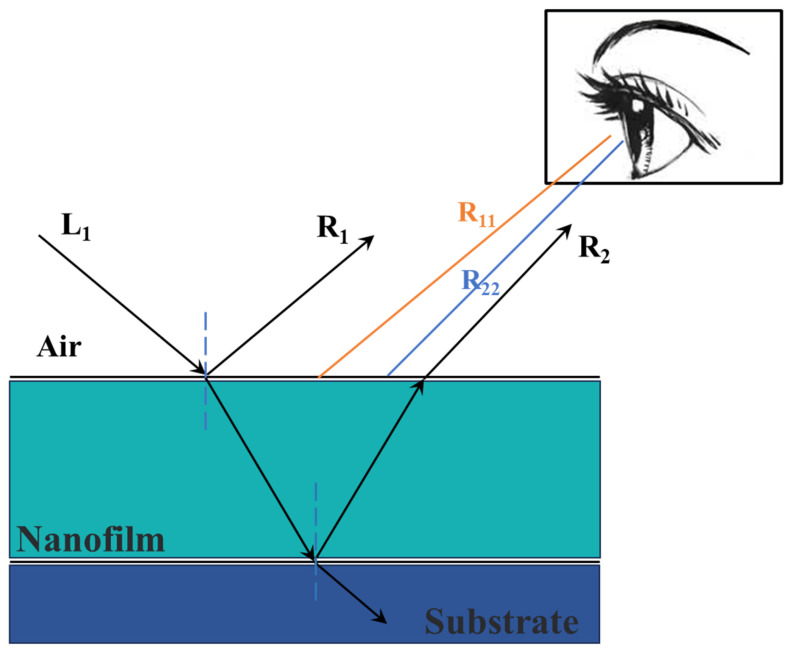
Principle of structural color generation from thin-film interference.

**Figure 2 biomimetics-09-00015-f002:**
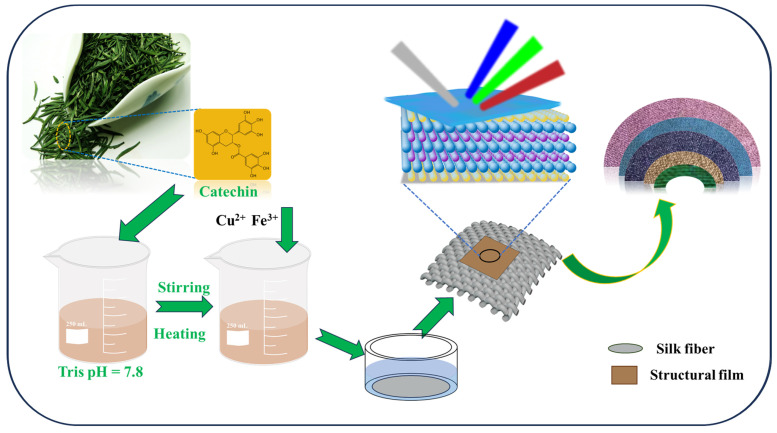
Preparation process of structural silk fiber.

**Figure 3 biomimetics-09-00015-f003:**
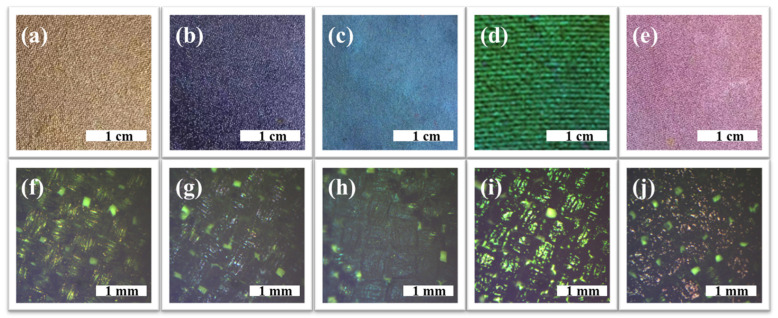
Photographs of structurally colored silk fabrics at different reaction times: (**a**) S-PCC-1.5 h, (**b**) S-PCC-2.5 h, (**c**) S-PCC-4 h, (**d**) S-PCC-6 h, and (**e**) S-PCC-9 h. Optical micrograph of structurally colored silk fabrics at different reaction times: (**f**) S-PCC-1.5 h, (**g**) S-PCC-2.5 h, (**h**) S-PCC-4 h, (**i**) S-PCC-6 h, and (**j**) S-PCC-9 h.

**Figure 4 biomimetics-09-00015-f004:**
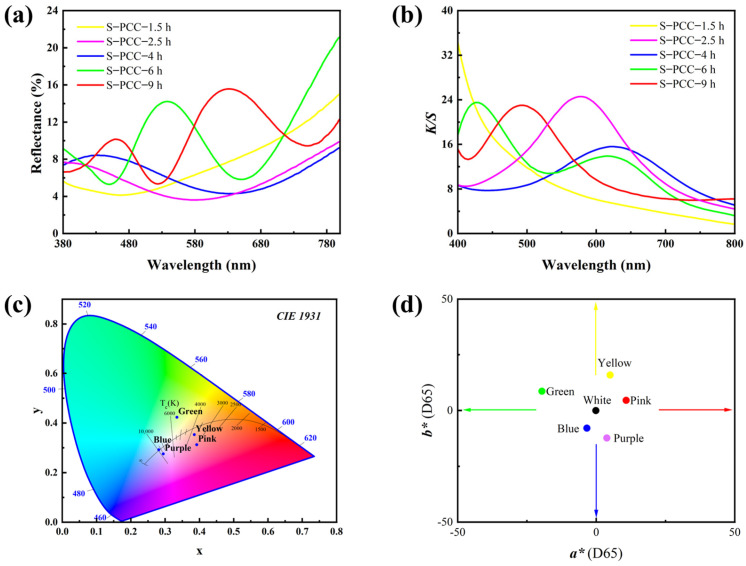
Optical properties of PCC and PCCmodified silk fabric: (**a**) reflectance spectrum of S−PCC-n; (**b**) K/S spectrum of S-PCC-n; (**c**) CIE1931 and (**d**) CIE1931 chromaticity coordinates of S-PCC-n (*** means CEI Color Specification System).

**Figure 5 biomimetics-09-00015-f005:**
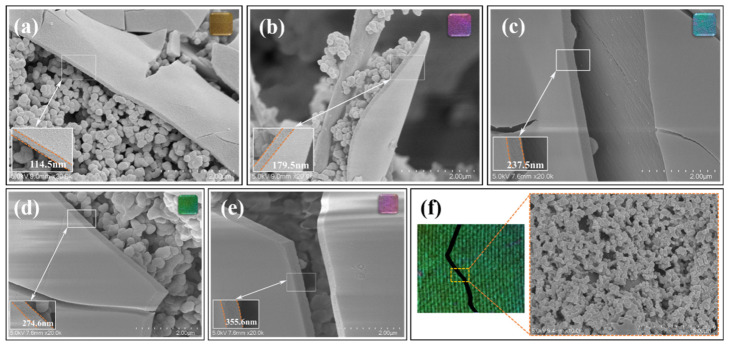
Cross-sectional morphology of a PCC structural color film with different polymerization time: (**a**) S-PCC-1.5 h (yellow), (**b**) S-PCC-2.5 h (purple), (**c**) C-PCC-4 h (blue), (**d**) S-PCC-6 h (green), and (**e**) S-PCC-9 h (pink); and (**f**) photograph and SEM image of S-PCC-4 h fabric with PCC partial-area film scraped off.

**Figure 6 biomimetics-09-00015-f006:**
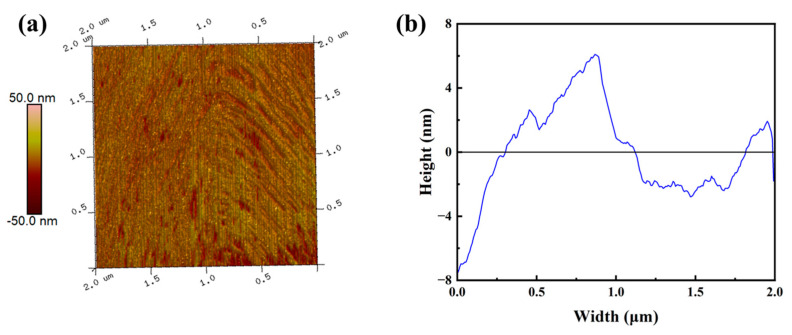
3D morphology AFM image (**a**) and surface height profile (**b**) of PCC-4 h film.

**Figure 7 biomimetics-09-00015-f007:**
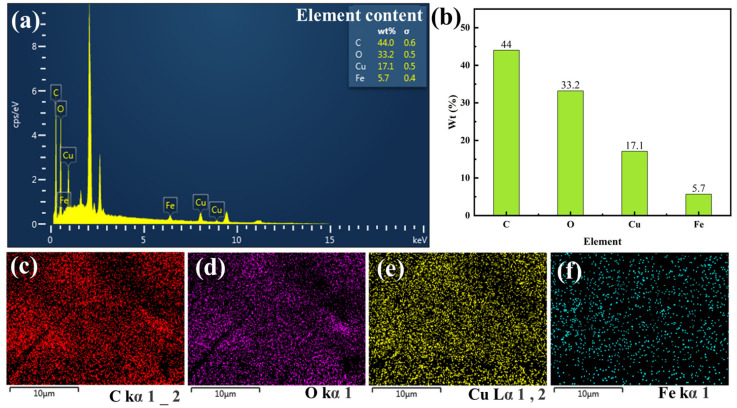
The surface of PCC-modified silk fabric: (**a**) EDS spectrum; (**b**) EDS element content; (**c**–**f**) EDS map distribution of the four elements C, O, Cu, and Fe.

**Figure 8 biomimetics-09-00015-f008:**
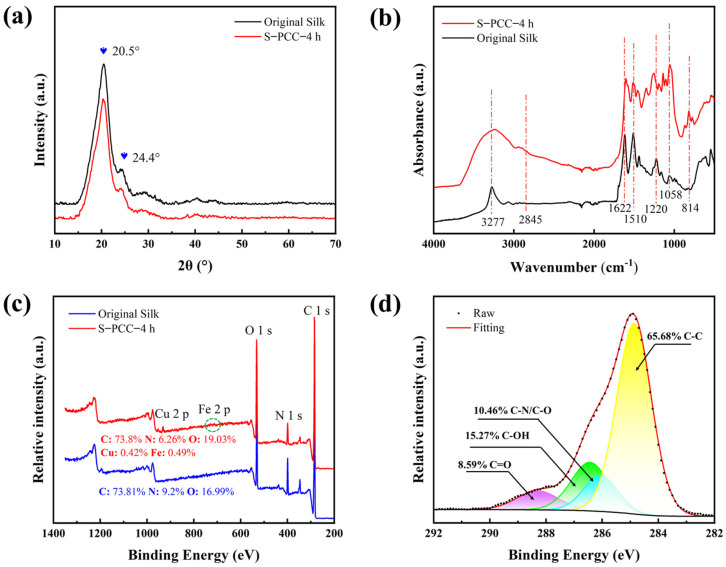
Surface chemical composition of original silk fabric and S-PCC-4 h: (**a**) XRD spectra; (**b**) FT-IR spectra; (**c**) XPS spectra; (**d**) high-resolution XPS spectrum of C1s.

**Figure 9 biomimetics-09-00015-f009:**
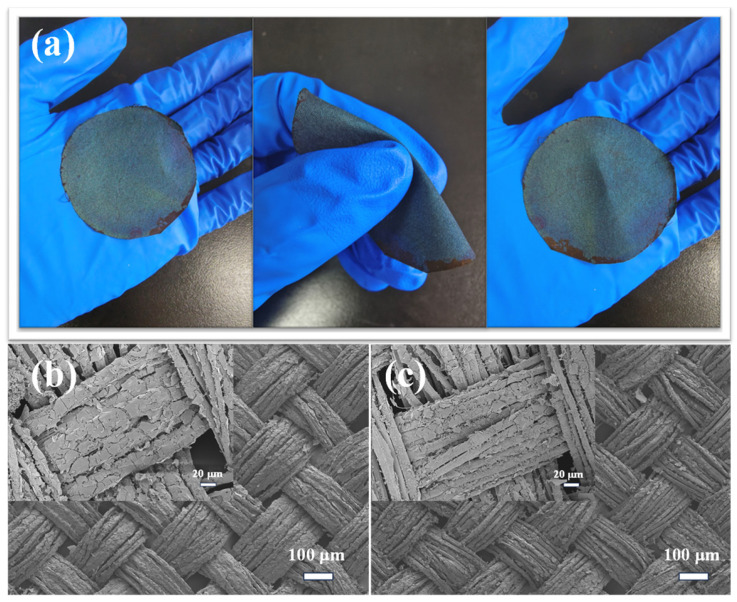
(**a**) Folding operation; (**b**) electron microscopic image of the structurally colored silk fabric before folding; (**c**) electron microscopic images of the crease of the folded structurally colored silk fabric.

**Figure 10 biomimetics-09-00015-f010:**
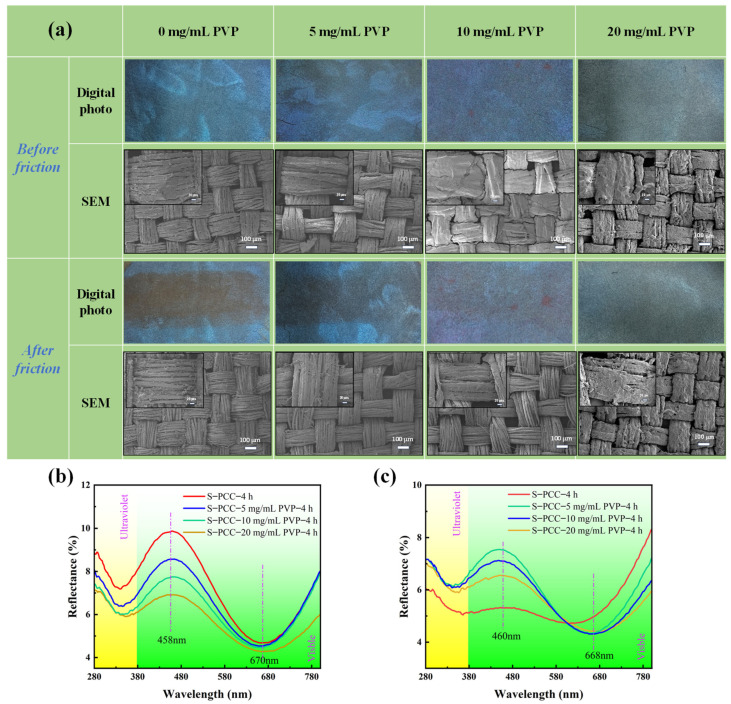
(**a**) Photographs and electron microscope images of S-PCC-4 h fabrics before and after friction with different amounts of PVP; and the reflectance spectra of PCC structural color silk fabric and PCC structural color silk fabric with different concentration of PVP: (**b**) before abrasion; (**c**) after abrasion.

**Figure 11 biomimetics-09-00015-f011:**
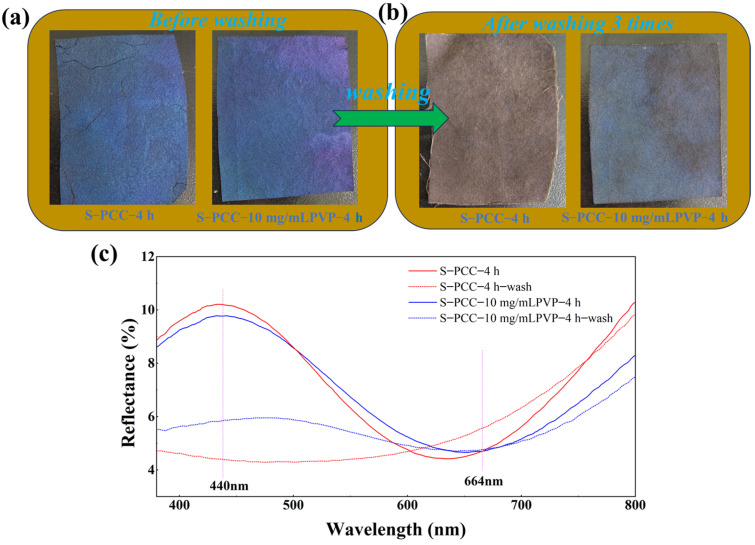
Photos of S-PCC-4 h and S-PCC-10 mg/mL PVP-4 h before soaping (**a**); after soaping (**b**); and reflection spectra of S-PCC-4 h and S-PCC-10 mg/mL PVP-4 h before and after soaping (**c**).

**Figure 12 biomimetics-09-00015-f012:**
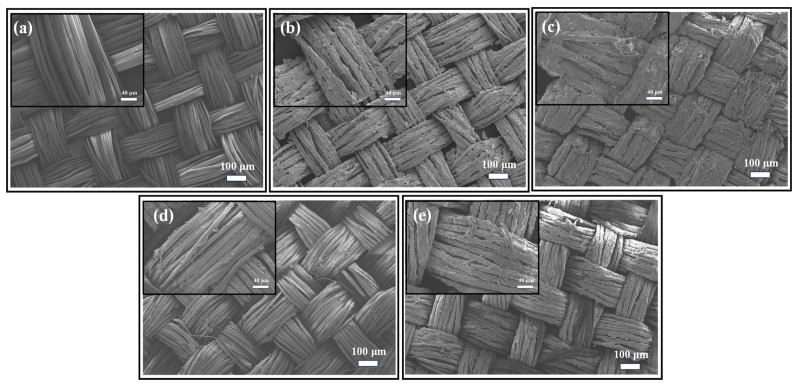
Electron microscopic images: (**a**) original silk fiber, (**b**) S-PCC-4 h before soaping (**c**) S-PCC-4 h after soaping, (**d**) S-PCC-10 mg/mLPVP-4 h before soaping, (**e**) S-PCC-10 mg/mLPVP-4 h after soaping.

**Figure 13 biomimetics-09-00015-f013:**
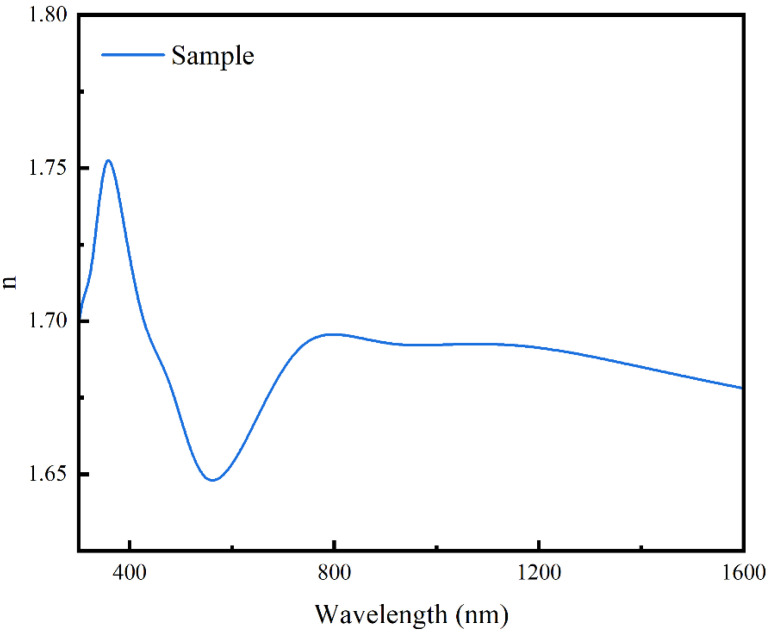
Real (n) parts of the complex RI (n*) of the PCC film. (n* is the refractive index divided into real and imaginary parts, and n is the real part).

**Figure 14 biomimetics-09-00015-f014:**
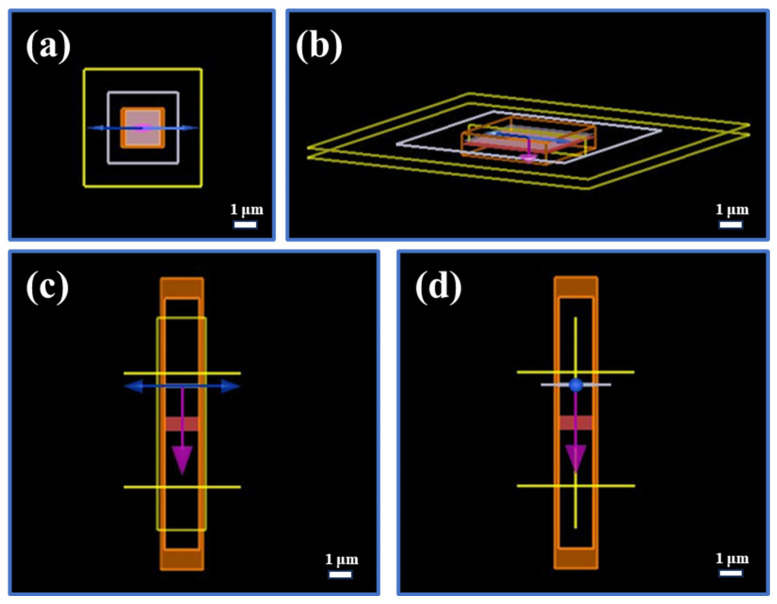
Different perspectives of the simulation model: (**a**) XY view; (**b**) perspective view; (**c**) XZ view; (**d**) YZ view.

**Figure 15 biomimetics-09-00015-f015:**
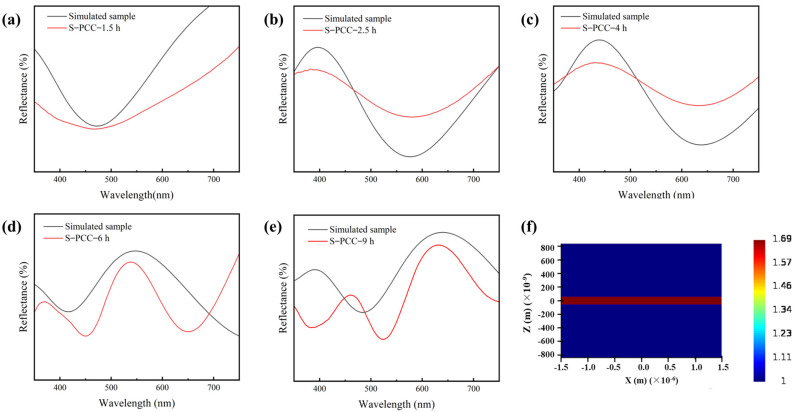
Analogue spectrum for PCC structural colored films. (**a**–**e**): Simulated spectra of PCC at different reaction times; (**f**) refractive indices in simulated space at 550 nm wavelength.

**Table 1 biomimetics-09-00015-t001:** Thickness of PCC upper film at different reaction times.

Reaction time (h)	1.5	2.5	4.0	6.0	9.0
Film thickness (nm)	114.5 ± 6.91	179.5 ± 5.82	237.5 ± 5.35	274.6 ± 9.82	355.6 ± 4.91

## Data Availability

The data presented in this study are available on request from the corresponding author.

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
