# Peer review of "Preparation of Natural Plant Polyphenol Catechin Film for Structural Coloration of Silk Fabrics"

_biomimetics, 2024, doi:10.3390/biomimetics9010015_

Round 1

Reviewer 1 Report

Comments and Suggestions for Authors

This manuscript expresses preparation of natural plant polyphenol catechin film structurally colored silk fabric. Before publication, there are some logic problems in the manuscript. Meanwhile, this manuscript is more like an experimental report than a research paper. Some more detailed explanations need to be mentioned.

-The abstract should be very clear and representative of the aim and main points achieved in the contribution. The readability and value of the abstract may be enhanced by a specific description of the better properties.

-The first line of paragraphs needs to be indented by two English characters.

-Lines 21-25, the sentences should be in the passive voice.

-What are the main principles of structural color, and why is film interference chosen?

-Lines 79-81, catechin also oxidize to a yellow-brown color in air, how do prove that it is due to metal cations?

-Why introduce Fe3+ when you already have Cu2+ as an oxidizing agent?

-In Fig.1, labels of various shapes need to be added. Meanwhile, does the green oval represent silk fiber? Does it change from green to brown? In addition, is the tea leaves added directly to the Tris without any other manipulation?

-Is there any cracking after the formation of the film?

-In Fig.3a and b, labels of the curves need to be changed to “S-PCC-?h”. Meanwhile, there should be an English space between the horizontal coordinate and the unit, and Figure 5b should be modified in the same way.

-Line 198-200, what effect does the refraction of light have on the structural color after a certain thickness is reached?

-In Fig.4a and c, cracks in the film are observed to be generated. What are the causes of this crack and how to avoid it?

-Oxidative polymerization mechanism of metal cations is ambiguous, which needs to be further stated.

-Table 1, it is too sample, and the standard deviation needs to be added.

-In mechanical stability testing, only abrasion test was measured.

-The conclusions should point out some important data.

-This study pointed out that PVP can slightly improve color fastness and fade after washing. What are the limitations of the application of PVP in textile dyeing and what are the further optimizations?

Comments on the Quality of English Language

English very difficult to understand/incomprehensible, much improvement must be needed in the revised manuscript.

Author Response

请参阅附件。

Reviewer 2 Report

Comments and Suggestions for Authors

In this article, the authors reported a method to make film with structured color. The procedure reported in this manuscript is relatively simple and the results are carefully characterized with SEM, XRD, FTIR, etc. However, there are some critical concerns about the results reported in this manuscript.

Major concerns:

1.       There is no evidence that the color observed by the authors is structure color. Structure color usually comes from nanoscale periodic features on the size comparable to light wavelengths. However, from the SEM images offered in Figure 4, there are no such nanoscale features in the film.

2.       Structure color shouldn’t be damaged easily via washing as shown in Figure 10. These results seem to suggest this is just a layer of chemical deposited on the silk.

3.       It is suggested that the authors should also confirm the microstructures change using SEM after washing.

Minor concerns:

4.       No scale bar available for the photos and micrographs included in Figure 2. They should be included.

5.       SEM images of the silk sample without treatment should also be included.

Comments on the Quality of English Language

English of this manuscript is fine.

Round 2

Reviewer 2 Report

Comments and Suggestions for Authors

Thanks for the authors careful consideration of my previous comments. I totally agree with the author that the color from the firm could come from interference. However, from the results and literature that the authors provided, it is not convincing that the color from the single layer of deposited film. Please refer to the Figure 3 in this review article:

Artificial Structural Colors and Applications

DOI: 10.1016/j.xinn.2021.100081

As it can be seen, it usually requires multilayer structures to have the distinguished color at different wavelengths unless the single materials have unique optical properties such as thin metal layer. In lines 208-217, the authors added new discussion about the origin of the interference color. However, these are only qualitative statement and need model support. It is suggested the authors should include theoretical or numerical model to compare the reflectance spectra of the film based on the thickness with experimental measurements.

Comments on the Quality of English Language

English is fine.

Round 3

Reviewer 2 Report

Comments and Suggestions for Authors

Thanks for the authors careful considerations of my previous comments and including of the new results. Based on my understanding, the new results have successfully proven that the color is not generated from structure of the materials. Instead, it is originated from the intrinsic refractive index modulation of the film as shown in figure 12, which is independent of the microstructure of the film. Thus, it is suggested that the authors should carefully change the terminology they used to describe their results.

Minor comments:

1. More details about the simulation should be included, such as the geometry and setting of the simulation model in FDTD.

2. Figure 13 should include a scale bar.

3. Figure 14f is unclear. Y axis is not labeled. Description shows wavelength dependence but there is no axis indicating wavelength.

Comments on the Quality of English Language

English is fine.

Round 4

Reviewer 2 Report

Comments and Suggestions for Authors

Thanks for the authors careful consideration of my previous comments. Here are my suggestions for clarification of the figures:

1. I believe Fig.14f is plotted for a specific wavelength, which is not mentioned.

Comments on the Quality of English Language

English is fine.
